# Outer-Membrane Vesicles of *Fusobacterium necrophorum*: A Proteomic, Lipidomic, and Functional Characterization

**DOI:** 10.3390/microorganisms11082082

**Published:** 2023-08-14

**Authors:** Prabha K. Bista, Deepti Pillai, Sanjeev K. Narayanan

**Affiliations:** 1Department of Comparative Pathobiology, Purdue University, West Lafayette, IN 47907, USA; pbista@purdue.edu (P.K.B.); pillai6@purdue.edu (D.P.); 2Indiana Animal Disease and Diagnostic Laboratory, Purdue University, West Lafayette, IN 47907, USA

**Keywords:** outer-membrane vesicles (OMVs), *Fusobacterium necrophorum*, proteomics, lipid profiling, cytotoxicity, outer-membrane proteins (OMPs), toxins

## Abstract

Outer-membrane vesicles (OMVs) are extruded nanostructures shed by Gram-negative bacteria, containing periplasmic contents, and often including virulence factors with immunogenic properties. To assess their potential for use in vaccine development, we purified OMVs from the *Fusobacterium necrophorum* subspecies *necrophorum*, an opportunistic necrotic infection-causing pathogen, and characterized these structures using proteomics, lipid-profiling analyses, and cytotoxicity assays. A proteomic analysis of density-gradient-purified *F. necrophorum* OMVs identified 342 proteins, a large proportion of which were outer-membrane proteins (OMPs), followed by cytoplasmic proteins, based on a subcellular-localization-prediction analysis. The OMPs and toxins were among the proteins with the highest intensity identified, including the 43-kDa-OMP-, OmpA-, and OmpH-family proteins, the cell-surface protein, the FadA adhesin protein, the leukotoxin-LktA-family filamentous adhesin, the N-terminal domain of hemagglutinin, and the OMP transport protein and assembly factor. A Western blot analysis confirmed the presence of several OMPs and toxins in the *F. necrophorum* OMVs. The lipid-profiling analysis revealed phospholipids, sphingolipids, and acetylcarnitine as the main lipid contents of OMVs. The lactate-dehydrogenase-cytotoxicity assays showed that the OMVs had a high degree of cytotoxicity against a bovine B-lymphocyte cell line (BL-3 cells). Thus, our data suggest the need for further studies to evaluate the ability of OMVs to induce immune responses and assess their vaccine potential in vivo.

## 1. Introduction

The Gram-negative anaerobic bacillus *Fusobacterium necrophorum* is an inhabitant of the respiratory, gastrointestinal, and genitourinary tracts of humans and cattle, with the ability to cause necrotic infections in both of its host species [1]. In humans, this opportunistic pathogen causes oropharyngeal infections, such as pharyngotonsillitis, peritonsillar abscess, Lemierre syndrome, and other acute forms of sore throat [2]. Conversely, *F. necrophorum* is the primary causative agent for a range of conditions in cattle, including liver abscesses, foot rot, foot abscesses, a necrotic laryngitis in younger calves known as necrobacillosis, or calf diphtheria in feedlot cattle, and metritis in dairy cattle [3,4,5].

With the dawn of the antibiotic era, disease-management strategies for *F. necrophorum* have significantly improved, and several antibiotics, such as tylosin, virginiamycin, and tetracycline, are effective against this organism [6]. However, in response to recent studies reporting the development of antibiotic resistance, the use of antibiotics as feed additives has been limited to prevent the further spread of resistance [7]. As alternatives, vaccines are attractive prophylaxes for *F. necrophorum,* and several approaches have been employed to identify vaccine candidates that are effective against fusobacterial infections [8,9]. In particular, vaccines based on toxins, such as hemagglutinin and leukotoxin, as well as several outer-membrane proteins (OMPs) and other virulence factors, have been investigated; however, no effective and economically viable vaccine has been developed [10,11,12].

Outer-membrane vesicles (OMVs) are spherical membrane-enclosed buds filled with periplasmic contents, which are produced by the blebbing of the outer membrane in Gram-negative bacteria [13]. These particles range in size from 20 nm to 300 nm and are primarily composed of lipids, proteins, and various pathogen-associated molecular patterns, including lipoproteins, peptidoglycan, and nucleic acids, derived from the parent bacterium [14]. These OMVs have been extensively studied in several organisms and are thought to play critical roles in the bacterial stress response, inter- and intracellular communication, gene transfer, the delivery of bacterial content into host cells, and host immune modulation [15,16]. Recent studies have further shown that OMVs from numerous pathogenic bacteria, including *Haemophilus* species, *Neisseria meningitidis*, *Vibrio cholerae*, and *Bordetella pertussis*, are enriched in virulence factors, signaling molecules, and other bacterial-toxin components [17,18,19,20].

Given their lack of infectious or replicative properties, general stability, and immunogenic properties against epithelial and immune cells [21], OMVs possess numerous characteristics that make them well-suited for vaccine development. Indeed, three OMV vaccines developed for serogroup B of *N. meningitides*—VA-MENGOC-BC, MenBVac, and MeNZB—were shown to be 83%, 87%, and 69% effective, respectively, in young adults [22,23]. Similarly, an OMV-based vaccine against *B. pertussis* infections was reported to display a greater immunological effect than either the acellular or whole-cell-based pertussis vaccines, suggesting it may be an ideal candidate for the development of an improved vaccine against this pathogen [24]. In some cases, OMVs have also been bioengineered to contain heterologous antigen, thereby generating vesicles with increased vaccine potential [25,26].

Importantly, despite promising data suggesting the efficacy of OMV-based vaccines, for any Gram-negative pathogen, it is necessary to elucidate the biological constituents of its associated OMVs to better estimate their vaccine potential. Moreover, studies aimed at determining the cytotoxic effects of OMVs on immune cells can reveal their potential for toxicity and help to identify the concentration suitable for vaccine study [27]. In this study, to determine whether fusobacterial OMVs have utility as possible vaccine candidates, we isolated OMVs from a laboratory strain of the *F. necrophorum* subspecies *necrophorum* and profiled the associated protein and lipid contents. In addition, we assessed the toxicity of purified *F. necrophorum* OMVs in bovine B lymphocytes (BL-3 cells). Through these analyses, we aimed to determine whether *F. necrophorum* OMVs are involved in the packaging and delivery of virulence factors to host cells and to estimate the antigenic potential and toxicity of an OMV-based fusobacterial vaccine candidate.

## 2. Materials and Methods

### 2.1. Bacterial Strain and Cell Culture

*Fusobacterium necrophorum* subspecies *necrophorum* 8L1 was isolated from a bovine liver abscess at a slaughterhouse in Kansas, United States [28]. The bacteria were streaked on blood agar plates (Thermo fisher Scientific, Waltham, MA, USA) and incubated in an anaerobic chamber at 37 °C. Isolated bacterial colonies were confirmed by matrix-assisted laser desorption/ionization (MALDI) mass spectrometry (MS) and the RapID Ana II System (Remel Inc., Lenexa, KS, USA). The immortalized bovine B lymphocyte BL-3 cell line (CRL-8037; American Type Culture Collection, Manassas, VA, USA) was grown in Corning Eagle Minimum Essential Media (EMEM; Corning, Thermo Fisher Scientific), supplemented with 10% fetal bovine serum (FBS) with 2 mM L-glutamine (Gibco, Thermo Fisher Scientific), at 37 °C, in 5% CO_2_.

### 2.2. Isolation and Purification of OMVs from Bacterial Culture Supernatants

The production of OMVs is influenced by various factors, including growth conditions, stress factors, and growth phases of bacterial cultures. The amount of OMV production can vary in response to these factors. During the late log phase, cells are known to produce a large quantity of OMVs. Hence, OMVs were isolated from late-log-phase bacterial-culture supernatants, as described by Kohl et al., with few modifications, as represented graphically in Figure 1A [29]. Briefly, bacterial isolates were grown on blood-agar plates; isolated colonies were inoculated in pre-reduced anaerobically sterilized brain heart infusion (PRAS-BHI) broth and grown in an anaerobic chamber at 37 °C until they reached late-log phase, corresponding to an optical density at 600 nm (OD_600_) of 0.8–0.9. Bacterial cultures were then centrifuged at 4000 rpm for 20 min at 4 °C to pellet the bacterial cells. Residual cells were removed by filtering the supernatants through a 0.2 μm vacuum filter, and the filtrates were concentrated using 100 kDa Amicon centrifugal filter units (MilliporeSigma, Burlington, MA, USA). Secreted OMVs were collected from the retentate by ultracentrifugation at 213,000× *g* for 4 h at 4 °C, using a SW40Ti rotor (Beckman Coulter, Brea, CA, USA). The OMV pellets were resuspended in 500 μL of saline (0.9% NaCl), and protein concentrations were determined by the Pierce Coomassie Plus (Bradford) Assay Kit (Thermo Fisher Scientific). The OMVs with protein concentrations > 1 mg were purified using OptiPrep (Iodixanol) Density Gradient Medium (Sigma–Aldrich, MilliporeSigma). In brief, different concentrations of OptiPrep were sequentially added to the tube, as follows: 2 mL (45%), 2.5 mL (35%), 2.5 mL (30%), 2.5 mL (25%), and 2.5 mL (20%), as shown in Figure 1B. Finally, 500 μL of isolated OMVs and normal saline (to fill the tube) were loaded on top of the gradient, and the tubes were subjected to ultracentrifugation at 150,000× *g* for 16 h at 4 °C. The different gradient layers were then collected, as described in [30], and each layer was ultracentrifuged at 150,000× *g* for 4 h at 4 °C; all pellets were resuspended in saline.

### 2.3. OMV Extraction in Iron-Deficient Conditions

To assess OMV production in response to limited iron availability, bacteria were cultured in iron-deficient media, prepared using Chelex resin (50–100 mesh; C7901, Sigma–Aldrich, MilliporeSigma, Rockville, MD, USA). All glassware used in these experiments was soaked in 2% nitric acid for 4–7 days at room temperature (RT) to remove iron and then rinsed with ultrapure water. Briefly, 15 g Chelex 100 resin was added to 300 mL of BHI and stirred for 1 h at RT, after which the resin was removed by filtration. The procedure was repeated twice, and then the iron-limited medium was supplemented with calcium and magnesium ions by adding 6 mg CaCl_2_·2H_2_O and 2.2 mg MgSO_4_ per 100 mL media. Iron concentration in the medium was adjusted to 42.1 μM Fe^3+^ by adding ferric chloride solution (FeCl_3_·6H_2_O). The medium was then boiled, supplemented with cysteine HCl (0.5 g/L), and anaerobically sterilized. Bacteria were inoculated and grown in iron-deficient media, and OMVs were isolated from 300 mL of bacterial culture supernatant, as described above for enriched PRAS-BHI media. Total protein concentrations in crude OMV pellets and purified OMVs were quantified using the Pierce Coomassie Plus (Bradford Assay Kit, ThermoFisher Scientific), according to the manufacturer’s instructions, as above.

### 2.4. Transmission-Electron Microscopy (TEM)

Resuspensions of each density-gradient layer were examined by TEM to identify those containing purified OMVs. In brief, each layer was resuspended in 2.5% glutaraldehyde and stored overnight at 4 °C. Samples were incubated on 400 mesh formvar-coated copper grids, rinsed with distilled water, and then negatively stained with 1% phosphotungstic acid. The images were acquired on an FEI Tecnai G^2^ F20 electron microscope equipped with a LaB_6_ source and operating at 200 kV (Purdue Electron Microscopy Facility, West Lafayette, IN, USA).

### 2.5. PCR Amplification of F. necrophorum-Specific Genes

The PCR used to assay for the presence of genetic material in OMV samples was performed using primers to the *F. necrophorum* genes encoding EUB (universal eubacterial primer), hemagglutinin (*haem*), and the 16s rRNA.

In addition, PCR amplification was performed on boil-prep DNA samples from *F. necrophorum* and OMV samples treated with and without DNase. The primers used for this analysis were as follows: E.U.B.: Forward: 5′ GGCTTAACACATGCAAGTCG 3′, Tm = 54.5 °C, Reverse: 5′ GGACTACCAGGGTATCTAATCCTG 3′; Tm = 55.8 °C; 16s rDNA: Forward: 5′ GAGAGAGCTTTGCGTCC 3′; Tm = 53.1 °C, Reverse: 5′ TGGGCGCTGAGGTTCGAC 3′; Tm = 60.6 °C; and *haem* gene: Forward: 5′ CATTGGGTTGGATAACGACTCCTA 3′; Tm = 57.1 °C, Reverse: 5′ CAATTCTTTGTCTAAGATGGAAGCGG 3′; Tm = 56.2 °C.

### 2.6. Proteomic Analysis of OMVs

Protein components were identified by reverse-phase liquid chromatography–electrospray ionization tandem MS (LC-ESI-MS/MS) using the Dionex UltiMate 3000 RSLC nano system, coupled to an Orbitrap Fusion Lumos Tribrid Mass Spectrometer (Thermo Fisher Scientific). Briefly, sample protein content was precipitated by adding 4 volumes of cold acetone, followed by overnight incubation at −20 °C. Precipitated samples were then centrifuged, and the dry protein pellets were dissolved in 50 μL of 8 M urea. The samples were then reduced with 10 mM dithiothreitol, alkylated with 20 mM iodoacetamide, and digested with a mixture of MS-grade trypsin and Lys-C (Promega, Madison, WI, USA), in a 1:50 ratio of enzyme to substrate, overnight, at 37 °C. The resulting peptides were desalted using Pierce C18 Spin Columns (Thermo Fisher Scientific), eluted with 80% acetonitrile (ACN) and 0.1% formic acid (FA), and dried at room temperature (RT) in a vacuum concentrator. Clean, dry peptides were resuspended in 97% purified water, 3% ACN, and 0.1% FA to a final concentration of 0.5 µg/µL, and 2 µL was used for LC-MS/MS analysis. Reverse-phase peptide separation was performed using a trap column (300 μm inner diameter (I.D.) × 5 mm long), packed with 5 μm of 100 Å PepMap C18 medium, after which peptides were separated on a reverse-phase column (75 µm inner diameter × 50 cm long), packed with 2 µm of 100 Å PepMap C18 silica (Thermo Fisher Scientific). The column temperature was maintained at 50 °C.

Mobile-phase solvent A contained 0.1% FA in water, solvent B contained 0.1% FA in 80% ACN, and the loading buffer contained 98% water, 2% ACN, and 0.1% FA The peptides were separated via reverse phase by applying them to the trap column in loading buffer for 5 min at a flow rate of 5 µL/min and eluting them with an 82-min linear gradient of 6.5–27% buffer B, followed by 40% of buffer B at 90 min and 100% buffer B at for 97 min. At this point, the gradient was held for 7 min before reverting to 2% buffer B at 104 min. The peptides were separated from the analytical column at a flow rate of 300 nL/min. The mass spectrometer was operated in positive-ion and standard data-dependent acquisition modes, with the advanced peak-detection function activated. Fragmentation of the precursor ion was accomplished by collision dissociation with higher energy at a normalized collision-energy setting of 30%. The resolution of the Orbitrap mass analyzer was set to 120,000 and 15,000 for MS1 and MS2, respectively, with maximum injection times of 50 ms for MS1 and 20 ms for MS2. Dynamic exclusion was set at 60 s to avoid repeated scanning of identical peptides, and the charge state was set at 2–7, with 2 as a default charge and a mass tolerance of 10 ppm for both high and low masses. The full-scan MS1 and MS2 spectra were collected in the mass ranges of 375–1500 m/z and 300–1250 m/z, respectively. The spray voltage was set at 2 kV, and the automatic gain control target was set at 4e^5^ for MS1 and 5e^4^ for MS2.

The LC-MS/MS data were analyzed using MaxQuant software v.1.6.3.3 [31] by comparing against the Uniprot *Fusobacterium necrophorum* protein database (www.uniprot.org). The cleavage enzymes were set as Trypsin/P and LysC, allowing up to two missed cleavages. The mass errors were set to 10 ppm and 20 ppm for MS1 and MS2, respectively. For fixed and variable modification, the alkylation of cysteine and oxidation of methionine were selected, respectively. The false-discovery-rate threshold was set at 0.01 for peptides and proteins.

### 2.7. Western-Blot Analysis

For Western blot analysis, 20 μg of total protein from OMV samples was separated by sodium dodecyl sulfate-polyacrylamide gel electrophoresis (SDS-PAGE) and then electrotransferred to polyvinylidene fluoride (PVDF) membranes (162-0182, BioRad, Hercules, CA, USA). The membranes were blocked with 3% bovine serum albumin in phosphate-buffered saline, supplemented with 0.05% Tween 20 (PBS-T) for 1 h at RT, followed by incubation with the following primary antibodies with 1:1000 dilution at RT for 2 h: anti-leukotoxin rabbit polyclonal and mouse monoclonal antibodies [32], anti-OmpH, and anti-43 kDa OMP rabbit polyclonal antibodies. All antibodies were produced for this study from Cocalico Biologicals Inc. (Stevens, PA, USA) or Lampire Biologicals (Pipersville, PA, USA) and are available in the laboratory [32,33]. After washing with PBS-T, membranes were incubated with 1:4000 dilution of goat anti-rabbit IgG, alkaline phosphatase conjugate secondary antibodies (12-448; EMD Millipore, MilliporeSigma) at RT for 1 h. Protein signals were visualized using the premixed BCIP/NBT premixed chemiluminescence detection solution (B6404; MilliporeSigma).

### 2.8. Lipid Analysis

Lipids were extracted from OMVs using the Bligh-and-Dyer method [34]. In brief, 1 volume of each OMV sample resuspended in 100 μL of normal saline was mixed with a 1:1 chloroform/methanol solution (*v*/*v*) and incubated at RT for 15 min. Chloroform and distilled water were added to the mixture, and samples were centrifuged for 10 min at 16,000× *g*. Finally, the chloroform layer containing the lipids was collected and placed in a new microcentrifuge tube, dried under a stream of nitrogen, and stored at −80 °C until use. For MS analysis, the dried OMV lipid extract was diluted in injection solvent, containing a 6.65:3:0.35 mixture of acetonitrile/methanol/300 nM of ammonium acetate (*v*/*v*/*v*). Multiple-reaction monitoring (MRM)-based targeted MS analyses were performed, wherein samples were injected at a flow rate of 10 μL/min, and data were acquired on a triple-quadrupole mass spectrometer (6410 Agilent QQQ, (Agilent Technologies, Santa Clara, CA, USA)) [35]. A separate sample injection was performed to profile each class of lipids. The raw MRM MS data were processed using an in-house script, and the resulting ion-intensity values were exported to Microsoft Excel. The absolute ion intensity of all lipids was normalized using an internal standard, and the relative quantity of any lipid with a sample ion intensity greater than that of the blank was determined. The resulting relative ion intensity of each lipid represents the proportion of the total lipid classes. The limit of detection (LOD) for lipid profiling was defined at LOD > 3. A similar methodology was used to profile and determine the relative amounts of lipids in bacterial cell pellets.

### 2.9. Lactate Dehydrogenase (LDH) Cytotoxicity Assays

The OMV cytotoxicity against a bovine B lymphocyte cell line (BL-3 cells) was determined by the release of lactate dehydrogenase (LDH), using the CyQUANT LDH Cytotoxicity Assay (Thermo Fisher), according to the manufacturer’s instructions. Briefly, 10^5^ BL-3 cells in 100 μL of supplemented EMEM (2 mM L-glutamine and 10% F.B.S.) were seeded in 96-well tissue-culture plates. Cells were incubated overnight at 37 °C in a 5% CO_2_ incubator and then treated with or without 200 µg/mL of OMVs for 6 h. Positive control cells for determining maximum LDH activity were treated with 10 μL of 10× lysis buffer, containing reduced Triton X-100, and incubated for 45 min at 37 °C in 5% CO_2_. To measure spontaneous LDH release, cells were treated with 10 μL of sterile milli-Q water/well. After incubation, plates were centrifuged at 4000 rpm for 5 min, and 50 μL of each sample was transferred to another 96-well plate. The positive control provided in the kit was also added to a labeled well. We then added 50 μL of the reaction mixture to each well and incubated the plate at RT for 30 min, protected from light. Finally, 50 μL of stop solution was added to each well and mixed by gently tapping and avoiding bubbles. The experiment was performed in three biological and three technical replicates. The percentage cytotoxicity was determined by measuring the absorbance at 490 nm and 680 nm and then calculating LDH activity with the following formula: % cytotoxicity = [(treated LDH activity − spontaneous LDH activity)/(maximum LDH activity − spontaneous LDH activity)] × 100.

### 2.10. Statistical Analysis

All data were obtained from at least three independent experiments. Data were analyzed using GraphPad Prism v.8.0 for Windows (GraphPad Software). Statistical significance was determined using an unpaired two-tailed Student’s *t*-test, with *p* ≤ 0.05 considered statistically significant.

## 3. Results

### 3.1. Characterization and Purity of OMVs

The purity and structure of the OMVs isolated from *Fusobacterium necrophorum* subspecies *necrophorum* 8L1 by density-gradient ultracentrifugation were determined by a TEM analysis of 12 total fractions obtained from the different iodixanol gradient layers, which revealed the presence of OMVs in the 3rd–6th (F3–F6) fractions (Figure 1B). The vesicle sizes ranged from 20 nm to 200 nm (Figure 2A). The purified OMV samples were cultured on blood-agar plates under anaerobic conditions for 48 h, and in all cases, no colonies were observed.

As iron is an essential nutrient for bacterial growth, its availability can affect the expression of iron-uptake systems and genes in bacteria. Bacteria often upregulate the expression of virulence factors as a way of scavenging iron from the host environment. This could also lead to changes in the production and composition of OMV. We next determined whether *F. necrophorum* OMV production differed under iron-limiting conditions relative to enriched conditions. To this end, the OMVs were isolated from an equal volume of bacterial-culture supernatant from cells grown in an iron-limited medium and enriched media. The TEM analysis revealed that the OMVs generated under iron-limited conditions were smaller in size, ranging from 20 nm to 40 nm vs. 20 nm to 200 nm for those produced under enriched conditions (Figure 2B). However, the OMV pellets obtained from the bacteria grown under iron-limited conditions were larger than those collected from the cells grown in the enriched media. Accordingly, for the cells grown in an equal volume of media under identical culture conditions, we found that the total protein content was higher in the OMVs isolated under iron-deficient conditions than in enriched conditions, as shown in Figure 2C.

We then assayed the presence of bacterial DNA in the purified OMVs by performing a PCR with primers to specific *F. necrophorum* genes, including EUB, *haem*, and the 16s rRNA gene. The amplification of all three products was detected with boiled samples of *F. necrophorum*; however, no amplifications were observed with the purified OMV samples.

### 3.2. Proteomics Analysis of OMVs

The purified OMVs were subjected to a proteomic analysis using a reverse-phase LC-ESI-MS/MS system. The resulting MS/MS spectra of the peptides detected in the *F. necrophorum* OMVs were then searched against the *F. necrophorum* genome. A total of 342 proteins were identified from this analysis, and they are listed in Appendix A. The OMV-associated proteins primarily included OMPs, autotransporter domain-containing proteins, toxins, and metabolic enzymes; 23% were found to be uncharacterized. When the LC-MS/MS output was stratified based on the intensity; the 10 highest-ranking proteins included the following: 43-kDa OMP, filamentous hemagglutinin N-terminal domain-containing protein, OmpA-family protein, leukotoxin-LktA-family filamentous adhesin, cell-surface protein (CSP), OmpH-family protein, and the FadA adhesin protein (Table 1). Given that the LC-MS/MS intensity corresponds to the amount of protein present in the OMV sample, these represent the most abundant OMV components, which mainly include OMPs and *F. necrophorum* toxins, which are thought to have pathogenic and antigenic properties.

To further validate the enrichment of the OMV proteins identified by the LC-MS/MS analysis, we predicted the subcellular distribution of the 50 highest-intensity proteins. To this end, we used the UniPROT ID mapping tool (https://www.uniprot.org/id-mapping, accessed on 30 October 2022) to obtain Fasta sequences and customized the search output for each protein and the online web-server packages pSORTb (https://www.psort.org/psortb/, accessed on 30 October 2022) v.3.0.3 and Cell-PLoc v.2.0 (http://www.csbio.sjtu.edu.cn/bioinf/Cell-PLoc-2/, accessed on 30 October 2022) to predict the subcellular localization. The results this analysis suggested that, among the 50 highest-intensity proteins in the OMV proteome, twenty-five were OMPs, four were periplasmic, five were inner-member proteins, 11 were cytoplasmic, two were extracellular, and the locations of three were unknown. Moreover, the relative proportions of outer-membrane and cytoplasmic proteins among the 50 highest-intensity proteins was found to be substantially higher (Appendix A).

We then confirmed the presence of several OMP-associated toxins and OMPs by performing SDS-PAGE and Western blot analyses with monoclonal anti-LktA, polyclonal anti-LktA, anti-OmpH, and anti-43-kDa OMP antibodies. Protein bands of approximately 110 kDa, 17 kDa, and 43 kDa were observed for the LktA, OmpH, and 43-kDa OMP, respectively. Thus, the results of our Western-blot analysis further confirmed the presence of toxins and OMPs in the *F. necrophorum* OMV samples (Figure 3).

### 3.3. Lipidomics Analysis of OMVs

We further performed a lipidomics analysis on the total lipid extract from the *F. necrophorum* OMVs. The compositional analysis of the molecules identified by MS revealed several classes of lipid molecule, including the following: the phospholipids phosphatidylethanolamine (PE), phosphatidylcholine–sphingomyelin (PC–SM), phosphatidylglycerol (PG), phosphatidylinositol (PI), and phosphatidylserine (PS); acetyl-carnitine (AC); free fatty acids; diacylglycerol (DAG); ceramide; triacylglycerol; and cholesteryl esters (CE). The primary lipid constituents in the *F. necrophorum* OMVs were identified as PE (27%), AC (20%), PC–SM (16%), and PG (15%), as shown in Figure 4A. In contrast, a larger percentage of PE (46.07%) was detected within the whole bacterial lipid profile, followed by PG (24.72%) and DAG (7.87%), as shown in Figure 4B. These results reveal that the lipid profiles of OMVs differ from that of the parent bacteria and can potentially be modified to meet specific functional requirements.

### 3.4. Functional Characterization of F. necrophorum OMVs

To identify the potential cytotoxic effects of the OMVs on the host cells, we used bovine B lymphocyte BL3 cells. Moreover, given that OMVs are also known to modulate the host immune system to favor bacterial survival and pathogenesis, assessing the effect of OMVs on BL3 cells could further help to elucidate their immunomodulatory potential. We found that the treatment with 200 µg/mL OMVs for 6 h induced significant (>60%) cytotoxicity (*p* < 0.01) in the BL3 cells compared to the cells treated with saline control (Figure 5). These results indicate that OMVs are toxic to living host cells, including immune cells, an effect that may be mediated by virulence factors and toxins within OMVs. We further note that BL3 cells treated with 10 µL of supernatant (leukotoxin-containing) from a 9-h *F. necrophorum* subspecies *necrophorum* 8L1 bacterial culture showed no significant cytotoxic effects relative to the cells treated with saline. This could have been due to several reasons; one crucial reason could be that the supernatant from a 9-h culture may not have had sufficient leukotoxin because of the high level of proteolytic activity. Many previous studies showed that the late logarithmic phase (7-h culture) had the highest leukotoxoid activity, after which the activity was lost or decreased drastically. However, we needed to use a 9-h culture to increase the OMV extraction. Therefore, the use of purified leukotoxin would have been ideal.

## 4. Discussion

Necrotic fusobacterial infections, such as liver and foot abscesses, foot rot, and necrotic laryngitis, are commonly observed in feedlot cattle and, sometimes, in dairy cattle, resulting in substantial economic losses for the cattle industry. Given that OMVs are often immunogenic, with an abundance of surface antigens in a native conformation, the development of an effective OMV-based vaccine is one potential strategy for controlling these devastating infections. In the present study, to explore this possibility, we purified OMVs from *F. necrophorum* 8L1, a strain isolated from a bovine liver abscess. The densities of the purified OMVs were between 1.13 g/mL and 1.15 g/mL (>1.11 g/mL), similar to OMVs from other bacteria, such as *Capnocytophaga ochracea, Porphyromonas gingivalis, Treponema denticola,* and *Tannerella forsythia* [36,37]. The OMVs were analyzed by TEM, revealing particles ranging from 20–200 nm, which is within the usual range for OMVs released by most Gram-negative bacteria [38,39]. We then performed proteomic and lipid-profiling analyses, revealing the presence of several OMPs, toxins, phospholipids, sphingolipids, and acetylcarnitine, and showed that OMVs display cytotoxicity assays against a bovine immune-cell line. To our knowledge, this is the first study to describe OMVs from *F. necrophorum.*

Naturally secreted bacterial products, OMVs allow cells to interact with a diverse range of environmental conditions and may help to promote survival under stress conditions [40]. Numerous studies have further postulated that OMVs are produced by the stress response as both an offensive and a defensive mechanism [41]. Here, to better understand OMV production by *F. necrophorum* in response to cellular stress, we used a nutrient-limitation strategy, growing cells in iron-deficient media to induce oxidative stress [42]. Notably, we observed an increase in the production of OMVs with high protein contents in bacteria grown in the media treated with an iron chelator compared to the cells grown in the enriched media. These findings are consistent with a previous study showing that *F. necrophorum* produces increased levels of leukotoxin under limited-iron conditions [43], with similar results also observed in other pathogenic bacteria, including *B. pertussis*, *Bordetella bronchiseptica*, and *Mycobacterium tuberculosis* [44,45]. Previous studies further showed that iron limitation leads to the activation of numerous virulence genes in many pathogenic bacteria, including *N. meningitidis* [46]. In addition, the results of investigations on *Aeromonas salmonicida* suggest that OMVs produced from limited-iron stressed bacteria are involved in iron acquisition and may be suitable for inducing protective immune responses [47].

The cytotoxic and immunogenic properties of OMVs result from the bacterial protein and lipid components embedded within these particles, which include periplasmic contents and OMPs. Our proteomic analysis of the *F. necrophorum* OMVs identified OMPs (e.g., 43-kilodalton OMP, OmpH, OmpA, FadA, and CSP) and toxins (e.g., filamentous hemagglutinin and leukotoxin) among the primary components. The presence of leukotoxin, OmpH, and 43-kDa OMP within the OMV particles was further confirmed by the Western-blot analysis. In addition, we detected numerous proteins containing an autotransporter domain and several periplasmic, cytoplasmic, and intermembrane proteins and enzymes, including eight proteases. One protease, present at high levels in the *F. necrophorum* OMVs, was identified as an Omptin-family outer-membrane protease (Appendix A). Notably, prior studies showed that OMV-associated proteases may be packaged as cargo proteins to promote pathogenesis and systemic infections [48].

The subcellular localization analysis of the 50 highest-intensity proteins further revealed that the largest percentage was found in the outer membrane, followed by the cytoplasm and inner membrane. This high abundance of OMPs in the *F. necrophorum* OMV samples was in agreement with the results of most published studies [44,49]. Moreover, the presence of numerous cytoplasmic proteins, including dehydrogenases, enolase, acetyltransferase, and 30 S ribosomal protein, suggests that these molecules are enclosed in the outer membrane or periplasm as cargo proteins and can fulfill essential biological functions. These proteins with multitasking and multifunctional abilities in addition to their primary functions are referred to as “moonlight proteins” [50,51]. These moonlight proteins have been identified in the OMVs of other bacteria, such as *Shewanella vesiculosa* [52,53].

The presence of filamentous hemagglutinin, a transmembrane glycoprotein that is essential for attachment and penetration in other pathogenic bacteria, was similar to the findings in studies on *B. pertussis*-Tohama-strain OMVs and suggests that this protein may be required for the colonization of the host mucosa [54]. However, future studies are needed to confirm the function of hemagglutination in *F. necrophorum*. Additional abundant OMV proteins identified via proteomics analyses include those containing an autotransporter domain. Prior studies showed that proteins with this domain often possess virulence-associated functions, including roles in adhesion, invasion, toxicity, and host-cell interaction [21,55]. Similarly, we found that leukotoxin, an essential toxin in *F. necrophorum,* is present at high levels within OMVs, suggesting that these particles may be cytotoxic to host cells [56]. Given that leukotoxin is known to affect host leukocytes, we tested this possibility by performing cytotoxicity assays with a bovine B-lymphocyte cell line and observed significant OMV-mediated cytotoxicity. Overall, the results of our proteomics analysis indicate the presence of numerous immunogenic antigens within OMVs, suggesting that these particles may be able to induce strong protective immune responses against fusobacterial infections.

The OMVs are produced by budding from the outer membrane and are believed to contain lipid constituents similar to those in the parent bacterium. Here, we performed a lipid-profiling analysis to identify the lipids in the *F. necrophorum* OMVs and determine whether they are distinct from those found in whole bacteria. The results showed that the purified OMVs and *F. necrophorum* cells contained similar lipid types, but that these were present at varying levels. In particular, PE, a typical conical lipid that induces membrane curvature [57], was found to be the most abundant lipid in both the OMVs and the parent bacteria, although the total PE content in the bacterial membrane differed from that in the OMV membrane. This difference could have resulted from the asymmetrical accumulation or depletion of PE in the membrane, leading to distinct structural changes. However, the presence of abundant PE in both the bacterial membrane and the OMVs suggests that these changes ultimately resulted in the blebbing of the membrane to form OMVs [58].

The high abundance of phospholipids detected in the OMV particles suggests that they are derived from the bacterial outer membrane. Phospholipids play crucial roles in maintaining membrane integrity, protecting against osmotic stress, regulating motility, and facilitating the selective permeability of the outer membranes [58,59]. Moreover, although rare in bacteria, sphingolipid was observed in the *F. necrophorum* OMVs and bacterial cells. This group of lipids has been linked to cell fitness, stress resistance, bacterial homeostasis, and host colonization in many members of the *Bacteroides* family [60,61], with reported roles in host–commensal interactions and pathogenesis via host-immune-system induction. In addition, the overall heterogeneous lipid composition of OMVs supports their formation by compartmentalization and the remodeling of the bacterial outer membrane. In eukaryotes, changes in cell-membrane-lipid composition and expression have been associated with aging, as well as with apoptotic changes that signal the host immune system to remove/respond to host-cell senescence [62]. Such studies have not been performed on bacteria, and we postulate that the OMV-lipid signature allows bacteria to deliver certain unique virulence factors to host cells, thus influencing the eukaryotic cells in and around it in vivo and in vitro.

The composition of OMVs, consisting of proteins and lipids, presents a distinct advantage for their potential use as antigen carriers and as natural adjuvants in vaccine development. These OMVs carry antigenic outer-membrane proteins (OMPs) and toxins, which may be recognized by Toll-like receptors (TLRs) on immune cells. However further investigations, such as immune-response studies, cell-based assays, and cytokine-release studies, are warranted to fully establish their immunological properties. It is important to note that if any toxic effects are observed with OMV-based approaches, certain modifications and/or engineering may be necessary to mitigate them. Additionally, the heterogeneity of OMVs, as demonstrated by previous studies, can affect cell interactions and functionality. Therefore, careful consideration of these factors is essential for planning and conducting future experiments [63,64].

In summary, to our knowledge, this study provides the first characterization of *F. necrophorum* OMVs and identifies several toxins, OMPs, and lipids with possible roles in host-cell targeting and immunogenicity. Future in vivo studies are therefore needed to elucidate both the toxic and the immunogenic effects of OMVs and to determine their vaccine potential. Furthermore, detailed proteomic and lipidomic analyses of stress-induced OMVs can determine whether these contain molecules possess functions that are distinct from those of normal OMVs. Comparative proteomics studies assessing OMVs from other *F. necrophorum* subspecies may also have utility for identifying unique proteins expressed in subspecies-specific OMVs.

## Figures and Tables

**Figure 1 microorganisms-11-02082-f001:**
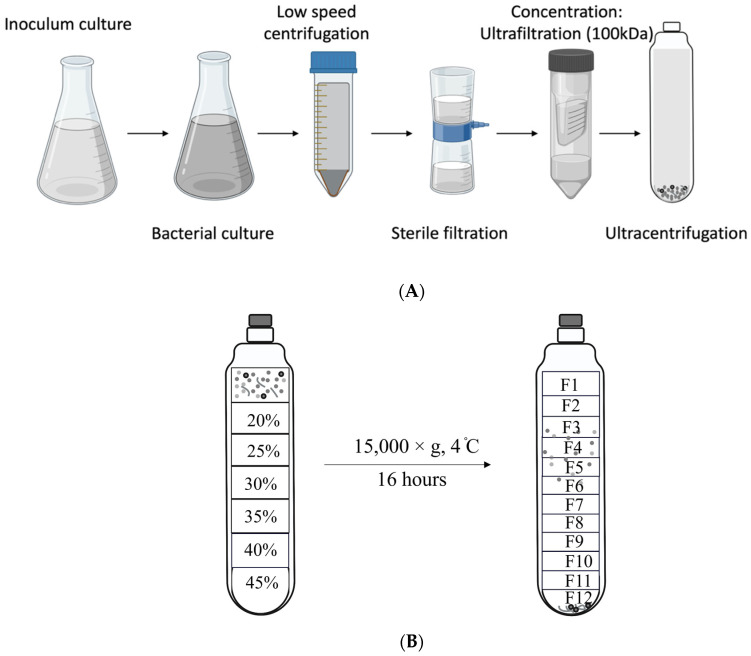
Schematic showing the experimental workflow for (**A**) outer-membrane vesicle (OMV) extraction and (**B**) purification using an OptiPrep (Iodixanol) Density Gradient.

**Figure 2 microorganisms-11-02082-f002:**
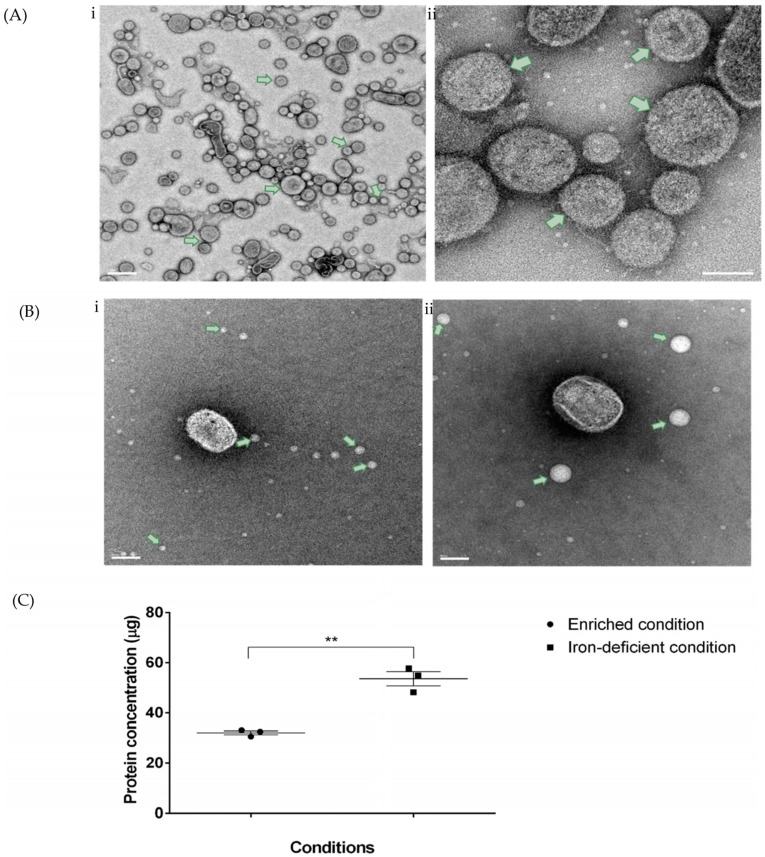
Characterization of purified OMVs from *F. necrophorum*.(**A**) Transmission-electron microscopy (TEM) images of density-gradient-purified OMVs from *F. necrophorum* (green arrows); (**i**) scale bar, 200 nm and (**ii**) 50 nm. (**B**) TEM images of OMVs produced from bacteria cultured under (**i**) iron-deficient and (**ii**) enriched conditions; scale bar, 50 nm. (**C**) Protein contents in crude OMVs produced from 300 mL of bacterial culture under enriched vs. iron-limited conditions. Significance was determined by unpaired, two-tailed Student’s *t*-test, ** *p* < 0.01. Bars indicate the mean ± standard error of the mean (SEM) of three data sets.

**Figure 3 microorganisms-11-02082-f003:**
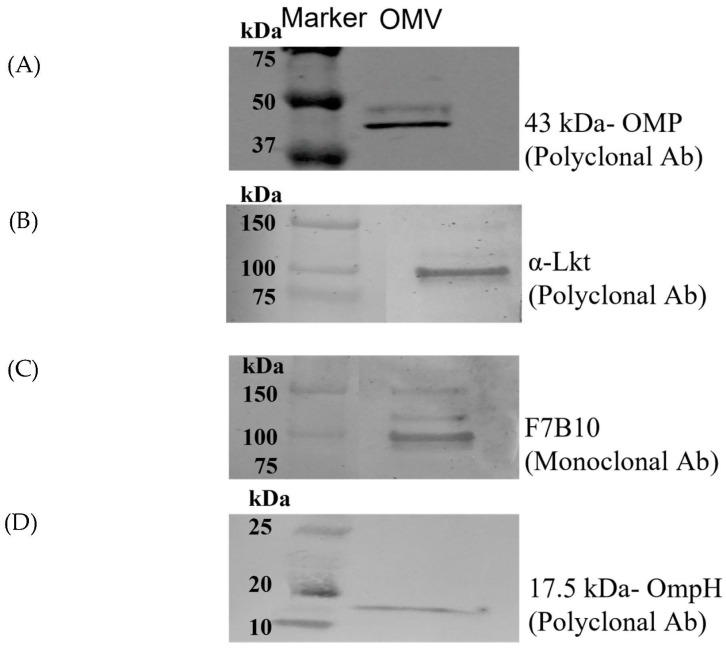
Immunoblot analysis using Western blotting to validate proteomic data. Purified OMVs were subjected to this analysis with the following antibodies: (**A**) 43-kDa OMP polyclonal antibodies, (**B**) polyclonal, and (**C**) monoclonal anti-leukotoxin LktA antibodies, as well as antibodies to (**D**) OmpH.

**Figure 4 microorganisms-11-02082-f004:**
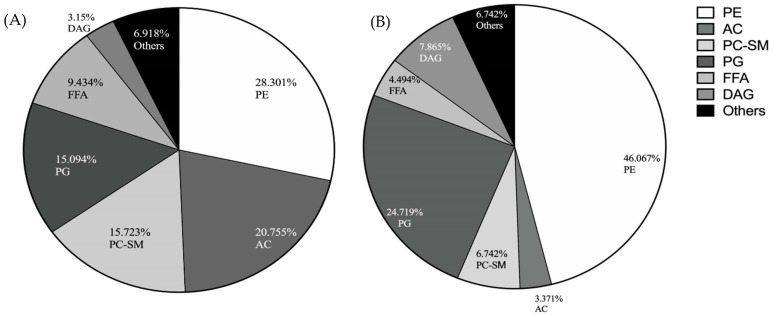
Lipid profiles of (**A**) purified OMVs and (**B**) whole *F. necrophorum* subsp. *necrophorum* bacterial cells. PE, phosphatidylethanolamine; AC, acetylcarnitine; PC–SM, phosphatidylcholine–sphingomyelin; PG, phosphatidylglycerol; FFA, free fatty acids; DAG, diacylglycerol. Others: ceramide, phosphatidylinositol, triacylglycerol, phosphatidylserine, cholesteryl esters.

**Figure 5 microorganisms-11-02082-f005:**
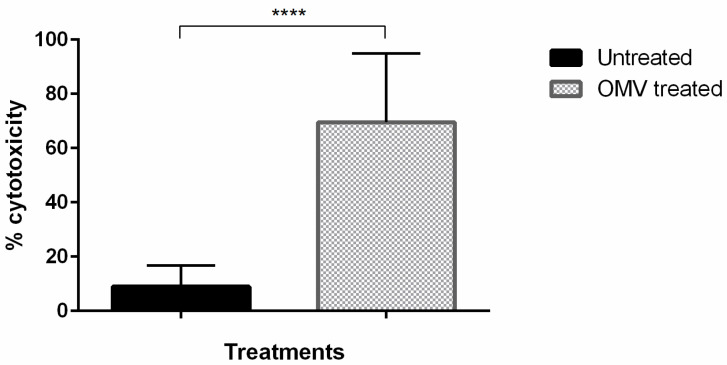
Cytotoxicity of OMVs against the BL-3 bovine B-lymphocyte cell line. Cytotoxicity was calculated based on release of lactate dehydrogenase from BL-3 cells treated with 200-μg/mL OMVs or control. Significance was determined by unpaired Student’s *t*-test, **** *p* < 0.0001. Bars indicate the mean ± SEM (standard error of the mean) of data from three experiments.

**Table 1 microorganisms-11-02082-t001:** Ten proteins with highest intensity identified in OMV samples by proteomics analysis.

Accession Number	F.A.S.T.A. Headers	Mol. Weight (kDa)	Intensity	Predicted Function
JQ740821.1	43-kilodalton outer-membrane protein	42.927	1.15 × 10^11^	Adhesion
AYZ72976.1	Outer-membrane-protein transport protein (aromatic-hydrocarbon-degradation protein)	51.728	4.40 × 10^10^	Cell envelope
AYZ73105.1	Filamentous protein containing the N-terminal domain of hemagglutinin	154.19	3.34 × 10^10^	Adhesion/carbohydrate-dependency hemagglutination activity
AYZ73911.1	Outer-membrane-protein assembly factor	79.432	2.02 × 10^10^	Protein translocase/membrane-protein-assembly factors
AYZ73504.1	OmpA-family protein	22.748	1.93 × 10^10^	Adhesion/porin
AYZ72978.1	Leukotoxin-LktA-family filamentous adhesin	335.96	1.85 × 10^10^	Toxin/adhesion
AYZ74730.1	Cell-surface protein (CSP)	66.384	1.54 × 10^10^	Adhesion/protein transport
AYZ74302.1	Complement-resistance protein TraT	25.823	1.36 × 10^10^	Outer-membrane-protein component/unproductive conjugation prevention
AYZ73912.1	OmpH-family outer-membrane protein	17.559	1.17 × 10^10^	Adhesion/periplasmic chaperone
AYZ73620.1	Adhesin protein FadA	14.319	1.10 × 10^10^	Adhesion

## Data Availability

The data presented in this study are available in the Appendix A.

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
