# Peer review of "Outer-Membrane Vesicles of Fusobacterium necrophorum: A Proteomic, Lipidomic, and Functional Characterization"

_microorganisms, 2023, doi:10.3390/microorganisms11082082_

Round 1

Reviewer 1 Report

The authors of this study conducted a comprehensive analysis of the outer membrane vesicles generated by Fusobacterium necrophorum subspecies necrophorum, focusing on their size, protein composition, and lipid content. Additionally, the authors aimed to assess the cytotoxicity associated with these vesicles. The goal of this research is to establish groundwork for leveraging these outer membranes in the development of a future vaccine.

This manuscript is well written and employs adequat methods. Nevertheless, the study would be enhanced by incorporating additional experimental conditions in several experiments. For instance, while the use of iron-deficient medium is exploited in only one experiment which does not bring much to the study as it stands, the idea is relevant particularly if the OMV would be further investigated for vaccine development.

comments:

L277-280: the authors show that under iron-deficient conditions. It would be interested to determine the ratio lipid/protein of the OMV produced in both normal and iron-deficient conditions. 

Furthermore, in L404, the authors mention that OMV produced in iron-deficient conditions in other species contain increased virulence factors due to their upregulation in such conditions. Based on this, it would be interesting to use the antibody Lkt that the authors used in fig3 to compare the abundance of one known virulence factor from F. necrophorum in such conditions.

Regarding the cytotoxicity effect, it would be interesting to add two conditions.
1. Investigating OMV from iron-deficient medium could provide insights. If those are enriched in virulence factors including the leukotoxin then these OMV might have a higher toxicity on this specific cell line. This should be tested.
2. Additionally, in fig5, the authors show significant cytotoxicity of the OMV that could not be reproduced with culture supernatant. However, it is important to note that the OMV number in supernatant is not similar to OMV prep quantified for proteins. This should be normalized, or alternatively a dose response should be tested.

L282, the authors used specific primers to detect DNA content in OMVs. Since the nucleic acids detected in OMV varies across species, it would be more conclusive to use a method to detect all nucleic acids such as using Qubit quantification.

Author Response

Reviewer 1

Please see the author’s response in the red font

Important Notice: The revisions made within the main text have been marked (highlighted) for your convenience. Please note that the line number referenced in this document may be subject to alteration.

The authors of this study conducted a comprehensive analysis of the outer membrane vesicles generated by Fusobacterium necrophorum subspecies necrophorum, focusing on their size, protein composition, and lipid content. Additionally, the authors aimed to assessthe cytotoxicity associated with these vesicles. The goal of this research is to establishgroundwork for leveraging these outer membranes in the development of a future vaccine.This manuscript is well written and employs adequat methods. Nevertheless, the studywould be enhanced by incorporating additional experimental conditions in severalexperiments. For instance, while the use of iron-deficient medium is exploited in only oneexperiment which does not bring much to the study as it stands, the idea is relevantparticularly if the OMV would be further investigated for vaccine development.

comments:

L277-280: the authors show that under iron-deficient conditions. It would be interested to determine the ratio lipid/protein of the OMV produced in both normal and iron-deficient conditions.

Complied with the reviewer’s comments, authors would like to clarify that in this particular study, only a limited quantity of OMVs were isolated and, the inclusion of iron-deficient conditions was specifically aimed at conducting basic experiments for the proteomic and lipidomic analyses of OMVs. Authors acknowledge and agree with the reviewer’s comments. They fully recognize the importance of conducting a detailed evaluation of iron- deficient conditions in future studies.

Furthermore, in L404, the authors mention that OMV produced in iron-deficient conditionsin other species contain increased virulence factors due to their upregulation in such conditions. Based on this, it would be interesting to use the antibody Lkt that the authors used in fig3 to compare the abundance of one known virulence factor from F. necrophorum in such conditions.

The authors appreciate the reviewer’s interesting suggestion. However, it is crucial to emphasis that comprehensive assessment of the iron deficient condition is necessary for our future studies. In this particular study, the production of OMVs under iron deficient condition resulted in minimal quantity, restricting its feasibility for conducting comprehensive investigations.

Regarding the cytotoxicity effect, it would be interesting to add two conditions.

  1. Investigating OMV from iron-deficient medium could provide insights. If those are enriched in virulence factors including the leukotoxin then these OMV might have a higher toxicity on this specific cell line. This should be tested.

Complied with the reviewer’s comments, authors have taken note of the comments and suggestions and they fully intend to consider them for future detailed experiments involving OMVs derived from an iron deficient condition

  1. Additionally, in fig5, the authors show significant cytotoxicity of the OMV that could not be reproduced with culture supernatant. However, it is important to note that the OMV number in supernatant is not similar to OMV prep quantified for proteins. This should be normalized, or alternatively a dose response should be tested.

Complied with the reviewer’s comments, authors would like to clarify that the purpose of using supernatant in the present study was to utilize the leukotoxin released in the supernatant as a positive control, rather than comparing its effects with the OMV supernatant.

L282, the authors used specific primers to detect DNA content in OMVs. Since the nucleic acids detected in OMV varies across species, it would be more conclusive to use a method to detect all nucleic acids such as using Qubit quantification.

Authors appreciate the constructive suggestion; however, it should be noted that due to unavailability of the instrument in the laboratory facility as suggested by the reviewer, authors were constrained to use the PCR-based quantification methods.                                                                               

Reviewer 2 Report

The authors utilized proteomics and lipidomics analyses to identify proteins and lipids in the isolated OMVs from a subspecies necrophorum of F. necrophorum under the oxidative stress condition induced by iron-limited culture to discover biomolecules that change abundance under the conditions. The result of most abundant proteins from LC/MS with DDA mode was validated by western blot. This study should be helpful in vaccine development for alleviating the outcome of F. necrophorum infection in people and cattle because of the high level of cytotoxicity in immune cells upon exposure to OMV. However, the following points need to be addressed, and a lot of typos should be corrected in this manuscript.

Major:

1.       L44: ‘In particular, vaccines based on toxins, such as hemagglutinin and leukotoxin, as well as several outer membrane proteins (OMPs) and other virulence factors, have been investigated [10–12]’. The authors state that already several vaccines are available and OMPs/virulence factors have been established. It is not clear from the results and discussion, which (if any) of the newly identified lipids/proteins are useful for vaccine development or other medical purposes. Ideally, for some of them immune responsibility (e.g. by probing with serum from infected animals) should have been verified. Currently, the study is mostly descriptive which severely limits the novelty and impact of this study.

2.       L236: ’ with or without 200 µg/ml of OMVs for 6 h…’ Why choose the total mass of 20 µg for 6 h to treat BL-3 cells? Have you established optimal dose and incubation time for OMV? What is the biological relevance of 20 µg? The real level of OMV under psychophysical conditions in humans and cattle or how much does it deviate from it?

3.       L263: ‘Vesicle size ranged from 20 to 200 nm (Figure 2A)’ It seems to characterize the vesicle distribution of OMV by TEM only, is laborious and not exact. So, did you also use NTA or DLS to obtain a comprehensive characterization?

4.       Obviously, Figure 3A is an attached blot with selected data. Please provide in the attachment a picture of stained gel with lane comparison of OMV and bacterial lysate to see difference in protein content.

5.       L363: ’We further note that….’ It is not clear how many volumes of Fnn supernatant were used, whether concentrated or not, but only with these details (culture time,….) it can be understood and reproduced. Next, L370: ‘we had to use 9-hour…’ Compared with 7 hours, how many more OMVs were obtained? Also, cell physiology changes during stationary phase adaptation, which should influence composition of OMVs? Any supported data to verify the effect of OMVs is not influenced?

6.       Is there any data to confirm or reject the hypothesis that the OMV population is heterogeneous? Any subtypes or classifications? Are there any publications about this? At least this must be discussed.

7.       Further statements for α-Lkt and F7B10 in Figure 3 are needed, and the molecular weight of Lkt in Figure 3 is about 100 kDa while in Table 1 ‘Leukotoxin LktA family filamentous adhesin’ is 335.96 kDa, hwy?.

Minor:

1.       L95:’OMV were isolated from late-log-phase bacterial…’, in humans, different states of cell also influence the composition of extracellular vesicle, dose the different growth phase produce different OMVs? – Please add info, if OMV production only occurs at certain physiological conditions or always in Fnn.

2.       Many of your abbreviations contain dots, which deviates from common nomenclature. I encourage removing the dot in ALL ABBREVIATIONS in this manuscript where they deviate from standard.

3.       Figure 1A is not aligned with the L100 description of how to prepare density gradient centrifugation. Please check and revise.

4.       L143 should indent.

5.       L185:’the spray voltage was set at 2’ add a unit after 2.

6.       L200:’ anti-leukotoxin…’ please provide the detail of Ab (company, cat number…) respectively, L205:’ Goat Anti-Rabbit IgG, Alkaline Phosphatase Conjugate secondary antibodies (12-448…’ only the information of anti-rabbit but you also use the 1st Ab derived the source of the mouse, please provide it.

7.       L240, a blank space should be between 50 and μl.

8.       L260, remove the bracket of ‘Figure 1(B)’.

9.       L269, add the dot after OMV.

10.   L287, Figure 1A is not mentioned in the main text.

11.   Check all figures, make sure the words align with the main icon, and adjust the consistent position of the bar in each picture.

12.   Regarding the different compositions of OMV and bacteria in part 3.3, more discussion associated with the specific functions  based on the changes in lipids would be better.

13.   Adjust the space between L351 and L352.

14.   L387, ref. 38 and 39 should be in a bracket.

15.   L429, the dot after moonlight protein should be removed.

16.   L430, the dot after vesiculosa should be moved after [52,53].

17.   L433, ‘OMVS.’ should be ‘OMVs’.

18.   Figure 4, the size of chart pie should be adjusted for identical image size.

Many minor mistakes in manuscript, thorough proofreading is required

Author Response

Reviewer 2

Please see the author’s response in the red font

Important Notice: The revisions made within the main text have been marked (highlighted) for your convenience. Please note that the line number referenced in this document may be subject to alteration.

Research article of Bista, Pillai and Narayanan described the isolation of outer membrane vesicles of Fusobacterium necrophorum and their proteomic analysis. Moreover, they were focused on the use of OMV as a potential vaccine target, therefore the bovine cells were examined in lactate dehydrogenase cytotoxicity assay. The methods and results are well described, I have only very few minor comments:

L100, L120 - temperature for the centrifugation is missing

  • Complied with the reviewer’s comments, all the experiments were performed at 4 Ö¯C and has been mentioned in line 101.

L141 - what is 'E.U.B.?

  • EUB is the Universal Eubacterial primers and mentioned in line 144.

L143, L196, L203,232, L281 - please remove dots in DNA/SDS/LDH

  • The dots could have been added during editing and have been removed.

L200 and 201 and 204 - please add the working concentration of antibodies

  • Complied with the reviewer’s comments, the working concentration of antibodies are added in line 201 and 206.

L261/262 - this statement you can transfer to discussion, here is it not relevant

  • Complied with the reviewer’s comments, the information on density gradient of OMV has been moved in the discussion section; Line 386-389.

Figure 1A is not cited in the paper

  • The figure in question serves as a graphical representation of the methodology and is mentioned in line 96.

the immunoblot analysis to validate the proteomic data is not so relevant here, however, it will be maybe nice to arrange the data according to the intensity (43kDa Omp as the first)

Complied to reviewer’s comment, the figure has been rearranged according to intensity observed in proteomics data, Line 335.

table A - please make the writing of Predicted function uniform

- Complied with the reviewer’s comments, the changes have been made in the predicted function column, in table 1, line 313

in the discussion, the vaccine potential is briefly mentioned, but maybe more information are necessary. The whole OMV wil be used or just single proteins/lipids/their combination will be more suitable?

- Complied to the reviewer’s comments, few sentences have been added in the discussion section, Line 473-483

L462/463 - I do not understand the sentence

- Complied with the reviewer’s comments, we have tried to simplify the sentence, Line 464- 466

Reviewer 3 Report

Research article of Bista, Pillai and Narayanan described the isolation of outer membrane vesicles of Fusobacterium necrophorum and their proteomic analysis. Moreover, they were focused on the use of OMV as a potential vaccine target, therefore the bovine cells were examined in lactate dehydrogenase cytotoxicity assay. The methods and results are well described, I have only very few minor comments:

L100, L120 - temperature for the centrifugation is missing

L141 - what is 'E.U.B.?

L143, L196, L203,232, L281 - please remove dots in DNA/SDS/LDH

L200 and 201 and 204 - please add the working concentration of antibodies

L261/262 - this statement you can transfer to discussion, here is it not relevant

Figure 1A is not cited in the paper

- the immunoblot analysis to validate the proteomic data is not so relevant here, however, it will be maybe nice to arrange the data according to the intensity (43kDa Omp as the first)

- table A - please make the writing of Predicted function uniform

- in the discussion, the vaccine potential is briefly mentioned, but maybe more information are necessary. The whole OMV wil be used  or just single proteins/lipids/their combination will be more suitable?

-L462/463 - I do not underand  the sentence

Author Response

Reviewer 3

Please see below for the author’s response in red font

Important Notice: The revisions made within the main text have been marked (highlighted) for your convenience. Please note that the line number referenced in this document may be subject to alteration.

 The authors utilized proteomics and lipidomics analyses to identify proteins and lipids in the isolated OMVs from a subspecies necrophorum of F. necrophorum under the oxidativestress condition induced by iron-limited culture to discover biomolecules that change abundance under the conditions. The result of most abundant proteins from LC/MS with DDA mode was validated by Western blot. This study should be helpful in vaccine development for alleviating the outcome of F. necrophorum infection in people and cattle because of the high level of cytotoxicity in immune cells upon exposure to OMV. However,the following points need to be addressed, and a lot of typos should be corrected in thismanuscript.

Major:

  1. L44: ‘In particular, vaccines based on toxins, such as hemagglutinin andleukotoxin, as well as several outer membrane proteins (OMPs) and other virulencefactors, have been investigated [10–12]’. The authors state that already several vaccinesare available and OMPs/virulence factors have been established. It is not clear from the results and discussion, which (if any) of the newly identified lipids/proteins are useful for vaccine development or other medical purposes. Ideally, for some of them immune responsibility (e.g. by probing with serum from infected animals) should have beenverified. Currently, the study is mostly descriptive which severely limits the novelty andimpact of this study.

In response to the reviewer’s comment, authors have conducted investigation on several virulence factors; however, none of them demonstrated efficacy as a standalone vaccine, leading to the unavailability of a vaccine at present. To enhance the clarity, modifications have been made to the text L46.

Considering that OMVs contain a comprehensive set of these virulence factors within a single entity, the authors believe that exploring the immune response elicited by OMVs is a promising avenue for investigation, rather than pursuing the development of a subunit vaccine using individual virulence factors. Importantly, no previous studies have examined OMVs of Fusobacterium necrophorum and the novel findings from proteomic analyses, revealing the presence of all these virulence factors in OMVs provide compelling evidence to support future studies.

  1. L236: ’ with or without 200 µg/ml of OMVs for 6 h…’ Why choose the total mass of 20 µg for 6 h to treat BL-3 cells? Have you established optimal dose and incubation time for OMV? What is the biological relevance of 20 µg? The real level of OMV under psychophysical conditions in humans and cattle or how much does it deviate from it?

In the present study, experiments were conducted using a series of concentrations ranging from of 0, 25, 50, 100, and 200 ug/mL in a single set (data not shown). The results demonstrated that the concentration of 200ug/ml induced cytotoxic effects on the host cell line. As a result, authors believe that using lower concentrations would be more appropriate for investigating the in-vitro immune response of OMVs.

  1. L263: ‘Vesicle size ranged from 20 to 200 nm (Figure 2A)’ It seems tocharacterize the vesicle distribution of OMV by TEM only, is laborious and not exact.So, did you also use NTA or DLS to obtain a comprehensive characterization?

Authors acknowledge that the techniques like NTA or DLS are employed for vesicle characterization. However, they would like to clarify that the unavailability of this equipment in the laboratory/ core facilities of their college led them to choose TEM as the method for this study. It is important to note that TEM has been widely accepted and utilized in numerous OMV studies for the purpose of characterization. 

  1. Obviously, Figure 3A is an attached blot with selected data. Please provide in theattachment a picture of stained gel with lane comparison of OMV and bacterial lysate tosee difference in protein content.

Complied to reviewer’s comment, authors acknowledge the concern raised by the reviewer.  The authors want to assure the reviewer that they have indeed conducted Western blot analyses using lysates and OMVs separately. However, they made a deliberate choice not to include lysate as a control alongside OMVs in the same blot. This decision was motivated by their primary objective, which focus on confirming the presence of specific proteins in the OMV samples.

- 5. L363: ’We further note that….’ It is not clear how many volumes of Fnnsupernatant were used, whether concentrated or not, but only with these details (culturetime,….) it can be understood and reproduced. Next, L370: ‘we had to use 9-hour…’Compared with 7 hours, how many more OMVs were obtained? Also, cell physiologychanges during stationary phase adaptation, which should influence composition ofOMVs? Any supported data to verify the effect of OMVs is not influenced?

In this study, a 10ul volume of a 9-hr supernatant culture of F. necrophorum was utilized as the source of leukotoxin. The volume information has been added in the main text L 365.

The authors chose 10ul volume to maintain equal volume as a control and clarify that it was not concentrated. This approach aligns with a study conducted by Pillai et al., in 2021, where a similar method was employed for leukotoxin analysis. Furthermore, the authors highlight that the optimal extraction of OMVs occurred during the late log phase (OD600 – 0.8-0.9), as supported by numerous publications on the OMVs. Consequently, the 9-hr culture was employed for comparison purpose, while a comparison at the7-hr mark was not conducted in this study due to the lack of significant results and hence the corresponding data was not presented in this study.

  1. Is there any data to confirm or reject the hypothesis that the OMV population is heterogeneous? Any subtypes or classifications? Are there any publications about this?At least this must be discussed.

Yes, the OMVs population are heterogenous in shape and protein/toxin content. The information has been discussed in the discussion section Line 481-483.

  1. Further statements for α-Lkt and F7B10 in Figure 3 are needed, and the molecular weight of Lkt in Figure 3 is about 100 kDa while in Table 1 ‘Leukotoxin LktA family filamentous adhesin’ is 335.96 kDa, hwy?.

Fig 3 legend has been rephrased L335.The discrepancy in size, where the observed size is 100kDa instead of the expected 335.96kDa, is due to the fact that the monoclonal and polyclonal antibodies used in the study were specifically raised against truncated forms of the leukotoxin. In support of this, the authors have referenced a study (Ref 32, on L208) that provides detailed information on the development of these antibodies against leukotoxin, including their usage with truncated forms.

Minor:

  1. L95:’OMV were isolated from late-log-phase bacterial…’, in humans, different states of cell also influence the composition of extracellular vesicle, dose the different growth phase produce different OMVs? – Please add info, if OMV production only occurs at certain physiological conditions or always in Fnn.

- Complied to reviewer’s comment, authors made appropriate edits by adding few sentences to address the reviewers concern in L95-98. In several studies on OMV extraction, it has been consistently mentioned that optimal conditions for extraction occurs during late log phase. This finding has been supported by a similar study referenced in L100.

  1. Many of your abbreviations contain dots, which deviates from common nomenclature. I encourage removing the dot in ALL ABBREVIATIONS in this manuscript where they deviate from standard. 

- The dots could have been added during editing and have been corrected.

  1. Figure 1A is not aligned with the L100 description of how to prepare density gradient centrifugation. Please check and revise.

- Density gradient centrifugation is illustrated on Figure 1B and has been checked.

  1. L143 should indent.

- Complied with reviewer’s comment, authors  have revised to address the feedback provided.

  1. L185:’the spray voltage was set at 2’ add a unit after 2.

- The unit has been added in line 188.

  1. L200:’ anti-leukotoxin…’ please provide the detail of Ab (company, cat number…) respectively, L205:’ Goat Anti-Rabbit IgG, Alkaline Phosphatase Conjugate secondary antibodies (12-448…’ only the information of anti-rabbit but you also use the 1st Ab derived the source of the mouse, please provide it.

The antibodies against leukotoxin were developed by the authors in their laboratory and detailed information can found in the citation 32. Company name utilized for generating antibody has been mentioned in L206

  1. L240, a blank space should be between 50 and μl. 

- Complied to reviewer’s comment, authors  have revised to address the feedback provided. L 243

  1. L260, remove the bracket of ‘Figure 1(B)’.

Complied with the reviewer’s comment, authors have revised to address the feedback provided.

  1. L269, add the dot after OMV.

- Complied to reviewer’s comment, authors  have revised to address the feedback provided.

  1. L287, Figure 1A is not mentioned in the main text.

The concern raised by the reviewer has been acknowledged by the authors. The fig 1A is the graphic presentation of the methodology and has been mentioned in L 99.

  1. Check all figures, make sure the words align with the main icon, and adjust the consistent position of the bar in each picture.

- Complied to reviewer’s comment, authors  have made necessary amendments to rectify any possible errors that may have been present.

  1. Regarding the different compositions of OMV and bacteria in part 3.3, more discussion associated with the specific functions based on the changes in lipids would be better.

Complied to reviewer’s comment, authors have made revisions to address the feedback provided in the discussion section, specifically in lines L 465-468 and L474-490. Despite the existing knowledge regarding the significant role of lipids as structural components within OMVs and their function as an indicator of bacterial fitness under stress or other environments, the precise mechanism by which each lipid contributes to these roles remain unclear.

- 13. Adjust the space between L351 and L352.

- Complied to reviewer’s comment, authors  have revised to address the feedback provided.

  1. L387, ref. 38 and 39 should be in a bracket.

Complied to reviewer’s comment, authors  have revised to address the feedback provided.

  1. L429, the dot after moonlight protein should be removed.

Complied to reviewer’s comment, authors  have revised to address the feedback provided.

  1. L430, the dot after vesiculosa should be moved after [52,53].

- Complied to reviewer’s comment, authors have revised to address the feedback provided.

  1. L433, ‘OMVS.’ should be ‘OMVs’.

Complied to reviewer’s comment ,authors have revised to address the feedback provided.

  1. Figure 4, the size of chart pie should be adjusted for identical image size.

- Complied to reviewer’s comment, the figure size has been adjusted, Line 352
